# Virtual Screening of Alkaloid and Terpenoid Inhibitors of SMT Expressed in *Naegleria* sp.

**DOI:** 10.3390/molecules27175727

**Published:** 2022-09-05

**Authors:** Jason Abraham, Neha Chauhan, Supriyo Ray

**Affiliations:** 1Department of Natural Sciences, Bowie State University, 14000 Jericho Park Rd., Bowie, MD 20715, USA; 2Department of Chemistry & Biochemistry, University of Texas El Paso, 500 W. University Ave., El Paso, TX 79968, USA

**Keywords:** *Naegleria*, primary amoebic encephalitis, ERG6, brain eating amoeba, Sterol-24C Methyltransferase (SMT), ergosterol, drug discovery, ligands, Tanshinone 2 A, ADME

## Abstract

The pathogenic form of thermophilic *Naegleria* sp. i.e., *Naegleria fowleri*, also known as brain eating amoeba, causes primary amoebic encephalitis (PAM) with a >97% fatality rate. To date, there are no specific drugs identified to treat this disease specifically. The present antimicrobial combinatorial chemotherapy is hard on many patients, especially children. Interestingly, *Naegleria fowleri* has complex lipid biosynthesis pathways like other protists and also has a strong preference to utilize absorbed host lipids for generating energy. The ergosterol biosynthesis pathway provides a unique drug target opportunity, as some of the key enzymes involved in this pathway are absent in humans. Sterol 24-C Methyltransferase (SMT) is one such enzyme that is not found in humans. To select novel inhibitors for this enzyme, alkaloids and terpenoids inhibitors were screened and tested against two isozymes of SMT identified in *N. gruberi* (non-pathogenic) as well as its homolog found in yeast, i.e., ERG6. Five natural product derived inhibitors i.e., Cyclopamine, Chelerythrine, Berberine, Tanshinone 2A, and Catharanthine have been identified as potential drug candidates based on multiple criteria including binding affinity, ADME scores, absorption, and, most importantly, its ability to cross the blood brain barrier. This study provides multiple leads for future drug exploration against *Naegleria fowleri*.

## 1. Introduction

*Naegleria fowleri* is an aquatic, thermophilic amoeba found in warm fresh waters that feeds on Gram positive and negative bacteria, algae, and yeast. Among all other species of *Naegleria*, only *N. fowleri* is known to be pathogenic. When *N. fowleri* enters our body, especially through the nasal cavity, it causes primary amoebic encephalitis (PAM) with a slim chance of survival [1,2,3]

*N. fowleri* exists in three stages, depending on the environment. The three stages are (a) cyst, (b) flagellate and (c) trophozoite stage. The cyst stage is formed in the case of unfavorable conditions and can survive low temperatures. The trophozoite stage is the only reproductive and pathogenic stage that flourishes between temperatures ranging from 35 °C to 46 °C. During PAM, nerve cells are degraded in this stage. The flagellate stage is formed in the presence of low nutrition at an optimum temperature range between 25 °C to 37 °C [4]. 

*N. fowleri* enters through the nasal cavity to the olfactory bulbs by following the olfactory nerves through the cribriform plate. Children are at higher risk because of the increased porosity of cribriform plate [3,4,5]. *N. fowleri* can move using acetylcholine as a ligand to bind to cell surface GPCR receptor [4]. About 95–97% of infected cases are terminal after the onset of infection within three to seven days and all infections can be traced to contact with water and water-related activities [3,6,7,8]. The symptoms of infection are loss of smell, fever, stiff neck, headache, anorexia, vomiting, seizures, and death [3,8]. These symptoms are also common with bacterial meningitis and may appear within 2–15 days, depending on the virulence of the strain [4,9]. The amoeba can internalize surface- bound antibodies and release cytolytic enzymes such as acid hydrolases, phospholipases, neuraminidases, and cysteine proteases, causing the destruction of brain cells [10].

Presently, there is no effective life-saving treatment for the infection, and multiple drugs are being repurposed and tested against the pathogen [11]. Amphotericin B is an antifungal drug that targets the ergosterol synthesis pathway and is a repurposed drug to treat *N. fowleri* infection [12]. The drug is used synergistically with other drugs: rifampin, azithromycin, and miconazole. Amphotericin B has various side effects such as chills, fever and nephrotoxicity. Miltefosine is another drug that is used synergistically with amphotericin B [13]. Configurin is an antifungal drug that is FDA approved as an orphan drug for the treatment of *N. fowleri* infection [14]. Though these treatments exist, the survival rate is minimal and the effectiveness of these drugs is limited due to higher toxicity levels, especially in children. These drugs not only have serious side effects but are also very expensive and therefore largely unavailable to the less privileged [4,14,15]

Sterols are an important part of lipid rafts and are involved in controlling fluidity, cell permeability, endocytosis, vesicle trafficking, cell signaling, and virulence in single-celled organisms [16]. Sterol 24-C Methyltransferase (SMT) is an enzyme involved in ergosterol biosynthesis and could serve as an important drug target for *N. fowleri*. SMT is involved in lipid metabolism and the biosynthesis of sterols [16]. This enzyme plays a vital role in the synthesis of ergosterol, a lipid found in multiple different pathogens and plants. SMT could serve as an excellent target because mammals do not synthesize it. Most eukaryotes cannot synthesize sterols and obtain it from food. Surprisingly, higher plants express two SMT enzymes that are specific to two different substrates to produce either C-24 ethylated or C-24 methylated phytosterols. In *N. fowleri,* there is only one gene that codes for only single SMT that produces only methylated sterols and no ethylated sterols. In *Naegleria* sp., 31-norlanosterol is metabolized to 4α-methylcholesta-8,24-dienol, which is converted to 4α-methylcholesta-7,24-dienol which in turn is metabolized to cholesta-7,24-dienol. Cholesta-7,24-dienol is later metabolized to cholesta-5,7,24-trienol and then converted to an intermediate 24 (28) methylene by SMT. This intermediate is then metabolized to ergosta-5,7-dienol which then is converted to ergosterol. Inhibiting SMT prevents formation of the critical intermediate *Naegleria* sp. necessary to synthesize ergosterol. 

Previous studies had identified three drug targets to inhibit ergosterol biosynthesis in *N. gruberi* i.e., sterol 14α-demethylase, sterol Δ8-Δ7 isomerase and sterol-24 C methyltransferase [16]. There have been studies to explore inhibitors for sterol Δ8−Δ7 isomerase but there have not been any studies to explore inhibitors for sterol-24 C methyltransferase specific to *Naegleria sp*. In our study, we aimed to identify naturally occurring compounds and their derivatives which can inhibit sterol-24 C methyltransferase activity and in the process inhibit the ergosterol biosynthesis pathway in *Naegleria sp*.

## 2. Results

The enzyme structure is homologically similar to the fungal ortholog, i.e., yeast. SMT derived from yeast (ERG6) was studied together with those derived from *N. gruberi* as the SMT enzyme is already established, unlike *N. fowleri.* Another reason is that *N. gruberi* is non-pathogenic and its lipid biosynthesis is well studied and established [17]. Two SMT enzymes have been found in *N. gruberi* as previously reported in higher plants. In plants, the two SMT enzymes use two different substrates, i.e., either C-24 ethylated or C-24 methylated phytosterols. In *Naegleria* sp., only one substrate has been detected, i.e., C-24 methylated phytosterols as both the enzymes are coded by the same gene. Hence, we predict that the two enzymes form from *N. gruberi,* i.e., XM [(XP_002671982.1)] and XP [(XP_002680047.1)] are isozymes. Both the isozymes were tested in this study and only those inhibitors were selected that showed strong binding affinity with both the isozymes and at the same time satisfied all other required criteria.


**Protein Sequence Analysis:**


The drug discovery process for SMT was challenging as the crystal structures of neither ERG6 nor SMT from *N. gruberi* have been solved as of yet. Before developing the protein model, the amino acid sequences of ERG6 were compared with SMT derived from *N. gruberi* using Clustal Omega. XM has 40.69% similarity while XP has 41.33% when compared to ERG6 as shown in Figure 1a. XP and XM had 77.56% similarity as shown in Figure 1a. The NCBI CDD web tool was used to check for the presence of conserved domains in the protein and the analysis proved that the partial structures covered the enzymatic methyl transferase domain, especially the S-Adenosylmethionine-dependent methyltransferase domain and the Sterol C-Methyltransferase domain responsible for SMT’s enzymatic action. XP showed amino acids LEU117, GLY118, CYS119, GLY120, VAL121, MET122, GLY123, ASN140, ASN140, GLY167, ASP168, PHE169, and ILE189 to be the binding site for S-Adenosylmethionine (SAM). XM showed amino acids LEU120, GLY121, CYS122, GLY123, VAL124, MET125, GLY126, ASN143, ASN144, GLY170, ASP171, PHE172, and ILE191 to be the binding site for SAM. These regions are covered by the XM Swiss model. 

Amino acid residues for SAM domain and the sterol methyltransferase C- terminal domain are highlighted in Figure 1b. Amino acid residues binding s-adenosylmethionine are shown by the red lines in Figure 1b. Emboss Needle alignment and CLUSTALW for ERG6, XM, XP SMTs are given in Appendix A.


**Protein Structure Modeling & Analysis:**


Since there is moderate sequence similarity between XP and XM and ERG6, therefore ERG6 could not serve as a template to develop structural models for *N. gruberi*. Protein structures were built using Swiss-modelers for XM and XP. The Swiss-modeler structures for XM and XP were partial structures. XP truncated protein covers 256 of 362 amino acid residues, while the XM truncated protein covers 251 of 367 amino acid residues. The structure of XP begins at Lys104 and ends at Lys360. The XM truncated protein begins at Asp110 and ends at Arg360. The SMT protein structures for XM and XP were selected based on sequence coverage and GMQE scores for the protein models generated by the Swiss-Model. The models are evaluated based on two quality measurement factors, namely GMQE and QMEANDisCo. The scoring function is between 0 and 1 for both. GMQE is abbreviated as a global model quality estimate and the score depends on coverage and alignment of the model to the template sequence. An incomplete structure would lead the GMQE scores to drop drastically depending on the coverage of the model. GMQE and QMEANDisCo help obtain more reliable quality scores. QMEANDisCo scores are based on average per-residue QMEANDisCo global scores and are estimated for large sets of models. The error estimate is calculated on the root mean square difference between the QMEANDisCo global score and IDDT where models similar in size to the template are considered.

The model for XM had a GMQE score of 0.41 and a QMEANDisCo Global score of 0.56 ± 0.05. Crystal structure of methyltransferase derived from *Methanosarcina mazei* (PDB: 3MGG) was used as a template for XM. The sequence identity of the model for the template was 17.98%. The Molprobity score of the XM structure was 2.48 and a clash score of 13.95 with 2.01% Ramachandran outliers.

Crystal structure of N-methyltransferase (PMT-2) derived from *Caenorhabditis elegans* complexed with S-adenosyl homocysteine and phosphoethanolamine (PDB: 4INE) was used as a template for homology modelling of XP. The sequence identity of the model to the template was 22.37%. The model for XP had a GMQE score of 0.40 and the QMEANDisCo Global score is 0.53 ± 0.05. The Molprobity score of the XP structure was 1.93 and a clash score of 9.75 with 1.57% Ramachandran outliners. The Ramachandran Plot server analysis revealed that 1.9% or 5 amino acids were in the highly questionable zone, namely LEU106, GLY225, GLY246, GLY264, LYS317. This was resolved after the minimization step.

Most importantly, methyl transferase domains, especially the SAM domain and the sterol c-methyl transferase domain responsible for SMT’s enzymatic action, are covered in both the XP and XM Swiss model.

Both the structures were dehydrated and the ligands were removed prior to energy minimization and then analyzed using the Ramachandran plot. The plot analysis revealed that the partial structure of XP was 99% error free, having one amino acid in questionable zones while XM had one amino acid in a questionable zone. The templates used by the Swiss model for models XM and XP are high resolution X-Ray crystallography structures, which are preferred to averaged NMR structures. Figure 2b,c show the Ramachandran plots for XP and XM respectively.

Along with the Swiss-modeler, structural models for both XP and XM were studied using I-TASSER and PHYRE2. The protein structure predicted by PHYRE2 was checked using the Ramachandran plot server and a large error rate led to the rejection of that model. More than 8.8% or 32 of the amino acids were in questionable zones, namely, THR44, ASP45, VAL60, PHE80, HIS81, HIS86, SER90, LEU117, ASN175, ALA180, ASP197, ASP228, ILE241, LYS243, GLY246, VAL273, ASP277, ASN280, GLY292, THR296, THR301, SER302, VL308, ILE314, VAL321, THR324, GLN326, LYS344, THR345, PRO350, PRO359, and LEU360. The questionable amino acids were found in the sterol methyltransferase domain, and thus had to be rejected. The I-TASSER server generated the five best models, and model 1 was selected based on the C-score. The C-score is calculated between −5 and 2, where higher values correlate with a more confident structure. Model 1 had the highest C-score of −1. The structure when checked using the Ramachandran plot indicated more than 7.8% or 20 amino acids in questionable zones, namely, GLY15, VAL36, ASN37, ASN51, GLY143, ASP169, PRO180, ASN194, GLY216, VAL218, THR221, SER222, ALA225, GLY232, ALA233, GLU234, GLY235, GLY238, GLY239, and SER248. This model was rejected due to a high error rate.

Yeast SMT protein was retrieved through the Alphafold Database. Alphafold uses a neural network called CASP which predicts structures with the highest accuracy, close to that of experimental models. AI’s novel neural network architecture and training based on evolutionary, geometric and physical constraints ensures the accuracy of the predicted structure [18]. Yeast derived SMT protein also known as ERG6 was analyzed using the Ramachandran plot analyzer to identify amino acid positions that are error free. Figure 2a shows the Ramachandran plot for the ERG6 generated by Chimera. The analysis indicated that the structure after minimization was 99% error free, with one amino acid in a questionable zone.


**Structural Comparison of Protein Models**


Since XP and XM had 77.56% sequence similarity, while XM has 40.69% and XP has 41.33% when compared to ERG6, it would therefore be interesting to compare their structures. Structural comparison between XP and XM; (Figure 3a) and then XM with ERG6 and XP with ERG6 show major differences in structure (Figure 3b,c respectively). Due to major differences in structure, it was imperative that each protein be analyzed separately for drug interaction studies.


**Druggability Site Analysis of the Protein Structures**


The analysis of the binding sites also revealed the pockets for each protein and the nature of the pockets through the use of the online tool https://proteins.plus/ (accessed on 31 July 2022). The protein pockets and the amino acids involved in binding can help us make an estimate of how hydrophobic/hydrophilic, positively/negatively charged, space of the pocket and what type of ligands may actually bind to that specific protein. The best druggable site with the highest score is coded in orange, followed by purple, while the worst is green, and the amino acids involved are given in Figure 4a–i.

The Appendix A for each protein structure i.e., ERG6, XM and XP, provide details of the drug score that is calculated based on hydrogen bond donors and acceptors, possible hydrophobic interactions and the number of charged amino acids. Appendix A provides details on the amino acid residues residing in the orange, purple and blue color coded regions respectively for ERG6, XM and XP proteins.


**Drug Docking and Binding Analysis**


The virtual screening tool PyRx was used to identify compounds with good binding scores. The compounds were then checked for ADME and drug likeness properties using SwissADME [18]. The structures of the prepped proteins were docked with ligands into the druggable sites to identify best hits. The results of the drug screening are summarized in Table 1.


**Drug-likeness and ADME studies for the ligands:**


Most of the ligands screened did not qualify the criteria to become drugs. The majority of compounds fail to succeed due to their pharmacokinetic properties and lack of drug-likeness. A pharmacokinetics analysis was conducted to determine the efficiency of a molecule being absorbed, distributed, metabolized and excreted out of the body. Certain criteria have to be met to make a chemical compound a potential drug. Molecular weight, lipophilicity, hydrogen bond donors and acceptors are a set of criteria which is tested using Lipinski’s rule of five. A molecular weight below 500Da, below five hydrogen donors, below 10 hydrogen acceptors, and an octanol-water partition coefficient LogP below 5 would suggest that the molecule is orally bioavailable [19]. The majority of the ligands docked were eliminated due to violations of the pharmacokinetic and drug-likeness criteria.


**Virtual Screening to identify potential ligands for ERG6 & SMT (*N. gruberi*):**


For the yeast ERG6, most ligands gave an excellent docking score of above −5 kcal/mol to even −11.2kcal/mol, but could not proceed due to lack of drug-likeness and unfavorable ADME properties. A total of 30 ligands showed binding above −8 kcal/mol. For the *N. gruberi* SMT XP and XM proteins, the ligands gave a similarly good docking score with variation in binding scores among the three proteins.

A comparative study of the docked ligands was performed on the basis of binding affinity. The ranking of the ligands on the basis of binding affinity is shown in Table 1.

Though the compounds showed excellent binding affinities as mentioned in Table 1, these compounds cannot be used for experimental studies as most of them have low druggability and cannot cross the blood brain barrier. *N. fowleri* crosses the blood-brain-barrier, (BBB) and any drug designed to affect its growth needs to be BBB permeable as well [3].

After screening for compounds that cross BBB and have an excellent druggability score, the following compounds were selected which can be used for further wet lab studies. These compounds shown in Figure 5a–e were selected as they fulfilled all the criteria mentioned above. Their binding affinities are shown in Table 1, their ADME scores are listed in Table 2, and their drug likeness are given in Table 3 and Table 4.


**Analysis of Docking Interactions of Cyclopamine with yeast ERG6 & SMT (*N. gruberi*)**


The ligand cyclopamine is a steroidal alkaloid isolated from *Veratrum californicum*, known for its activity against various carcinomas [20]. Two derivatives of cyclopamine, vismodegib and sonidegib, have been approved by the FDA for cancer treatment [21,22]. The yeast SMT and cyclopamine had a binding affinity of −8kcal/mol, while for XM and XP it was −8.6, 10.3 Kcal/mol. For ERG6-cyclopamine, electron cloud interactions like alkyl and acceptor-acceptor are formed with the amino acids Ala101 and Glu136. Figure 6a depicts acceptor-acceptor interactions in red between Glu136 and ligand, alkyl interactions with Ala101 in orange. Figure 6b depicts yeast ERG6 protein surface hydrogen bond donor information. The methyl transferase domain (SAM dependent Methyl transferase family) domain ranges between amino acid 124 to 222, in ERG6 and the cyclopropane fatty-acyl-phospholipid synthase ranges from amino acid residue 66 to 277 in yeast. These domains have been targeted by the cyclopamine in ERG6 as shown in Figure 6a,b.

For XM-cyclopamine, electron cloud interactions like pi-alkyl, and alkyl interactions are formed with amino acid residues Trp223, Ile317, and Val321. Figure 7a depicts alkyl interactions in red, pink, and orange between the ligand and Val321, Ile317. Pi-alkyl interactions between ligand and Trp223. Figure 7b, depicts the XM protein surface and ligand. As mentioned earlier, the methyl transferase domain i.e., sterol methyltransferase C-terminal domain ranges from amino acid 300–362 in XM. This domain has been targeted by cyclopamine in case of the XM.

For XP-cyclopamine interactions such as pi-alkyl, alkyl interactions and hydrogen bonds are formed between amino acid residues Tyr218, Glu271, Ile275, Trp284, Trp298. Figure 8a, depicts pi-alkyl interactions between a ligand and Trp298, Trp284, Tyr218 in purple. Hydrogen bonds formed between the ligand and Glu278, Trp218, are shown in green. The alkyl interaction between the ligand and Ile275 is shown in purple. Figure 8b depicts the XP protein surface and ligand. As mentioned earlier, the methyl transferase domain (SAM dependent Methyltransferase family) ranges from amino acid 115–215 and the sterol methyltransferase C-terminal domain ranges from amino acid 298–358. The second domain was targeted by this ligand in the case of XP. The amino acids involved in the binding pockets are different for each protein. Though cyclopamine has good properties for further studies, it is known to inhibit the sonic hedgehog pathway. The compound can cross BBB and has good bioavailability.


**Analysis of Docking Interactions of Chelerythrine with yeast ERG6 & SMT (*N. gruberi*)**


The ligand Chelerythrine is a benzophenanthridine alkaloid isolated from the root of *Zanthoxylum simulans*. It acts as a protein kinase inhibitor and has antibacterial and anticancer properties [23,24]. Recent studies have also confirmed activity against COVID-19 and even acts as a neuromodulator and anti-inflammatory [25]. The binding affinity of chelerythrine with ERG6 was −7.9 kcal/mol, while for XM and XP it was −7.8 kcal/mol and −8.6 kcal/mol, respectively. For yeast-chelerythrine, electron cloud interactions like pi-alkyl, pi-sigma, conventional hydrogen bonds and alkyl interactions/bonds are formed with the amino acids Tyr83, Phe80, Leu31, Leu28, Val337, Val321, Met324, and Met320. Pi-alkyl interactions in red, yellow, magenta, and blue are depicted between the ligand and Val321, Met324, Leu31, and Phe80. Pi-sigma shown in purple depicts the interaction between ligand and Leu31, Val321, Met320, Leu28. Alkyl interactions in pink depict the interaction between Val337 and Val321. Conventional hydrogen bond interactions in green are depicted between the ligand and Tyr83; Figure 6c,d depicts the ERG6 protein surface and Chelerythrine. Chelerythrine targeted the Cyclopropane fatty-acyl-phospholipid synthase (amino acid 66–277) domain and the sterol methyltransferase C-terminal (amino acid 306–368) domain for ERG6. For XM-chelerythrine, electron cloud interactions like pi-alkyl, pi-pi and hydrogen bonds are formed with amino Pro127, Ile192, Phe134, Asp110. Figure 7c depicts alkyl interactions in pink between Pro127 and Ile192. The pi-pi interaction between ligand and Phe134 is shown in pink. Conventional hydrogen bonds between Asp110 and the ligand is shown in green. Figure 7d depicts XM SMT protein surface and chelerythrine. Chelerythrine targeted the methyltransferase (amino acid 118–219) domain and the mycolic acid cyclopropane synthase (amino acid 61–274) domain for XM protein. For XP-Chelerythrine, pi-alkyl, pi-anion, pi-pi T-shaped, pi-pi stacked, alkyl interactions are formed with amino acids Tyr218, Val273, Glu190, Trp284, Trp298, Phe353. Figure 8c, depicts pi-anion interactions between the ligand and Glu190 shown in golden yellow. Pi-pi T-shaped interactions between the ligand and Trp298, Trp284, and Tyr218 are shown in pink. The pi-alkyl interaction between the ligand and Phe353 is in purple. The alkyl interaction between the ligand and Val273 shown in purple. Figure 8d depicts the XP SMT protein surface and Chelerythrine. Chelerythrine targeted sterol methyltransferase C-terminal (amino acids 298–358) domain in the case of XP protein. The amino acids involved in each protein-ligand interaction are different compared to the others.


**Analysis of Docking Interactions of Berberine with yeast ERG6 & SMT (*N. gruberi*)**


Berberine is an isoquinoline alkaloid derivative and is present in Chinese herbs, Oregon grape, and berberry. It is known to modulate gene expression and induce cell arrest or apoptosis [26]. It has potential as an anticancer, antidiabetic, anti-inflammatory and can influence glucose and lipid metabolism [27]. It can also act as an inhibitor for acetylcholinesterase, tyrosine-phosphatase, choline-esterase, phospholipase and the potassium channel. It is also not linked to any instances of toxicity. It has good bioavailability and can cross the blood-brain-barrier.

The binding affinity of the compound with ERG6, XM, XP SMTs are −7.3 kcal/mol, −7.8 kcal/mol, −8.3 kcal/mol, respectively. For ERG6-berberine, electron cloud interactions like pi-alkyl, amide-pi stacked and alkyl interaction bonds are formed with amino acid residues Lys163, Tyr160, Lys159, Ala156, Lys175, Asp152, Ile22. Figure 6e depicts pi-alkyl interactions in brown and yellow between the ligand and Ala156. Amide-pi stacked interactions in pink between the ligand and Lys159. Alkyl interactions in brown, red, and orange between Lys175, Ile22, Lys163. Carbon hydrogen bond interactions in green between ligand and Asp152. Figure 6f depicts the ERG6 protein surface and Berberine. Berberine targeted the methyltransferase (amino acids 124–222) domain and the cyclopropane fatty-acyl-phospholipid synthase (amino acids 66–277) domain for ERG6. For XM-Berberine, electron cloud interactions like pi-alkyl, pi-pi stacked, pi-pi T-shaped alkyl interactions and hydrogen bonds are formed with amino acid residues Tyr190, Leu111, Asp110, Phe134, and Asn130. Figure 7e Alkyl interactions are shown in metallic pink between ligand and Leu111. Conventional hydrogen bond between ligand and Asp110, Asn130 in green. Pi-alkyl interactions are shown in metallic pink, pink between ligand and Ile131, Tyr190. Pi-Pi T-shaped interactions in pink between ligand and Tyr190, Ile131, Phe134. Conventional hydrogen bond interactions in green between ligand and Asp110, Asn130. Figure 7f depicts the XM protein surface and Berberine. Berberine targeted the methyltransferase (amino acids 118–219) domain and the mycolic acid cyclopropane synthetase (amino acids 61–274) domain for XM. For XP-Berberine, electron cloud interactions like pi-alkyl, pi-sigma, pi-pi stacked are formed with amino acids Tyr187, Ile189, Tyr218, Trp298, and Phe353. Figure 8e depicts the pi-alkyl interactions between the ligand and Tyr187, Try218, Phe353, Trp298, and Ile189. Pi-Pi stacked interactions between ligand and Tyr218 are shown in pink. Pi-Alkyl interactions between ligand and Trp298, Tyr218 in purple. Pi-Sigma interaction between ligand and Tyr187 is shown as light purple. Figure 8f depicts the XP SMT protein surface and Berberine. Berberine targeted Sterol methyltransferase C-terminal (amino acids 298–358) domain for XP SMT.


**Analysis of Docking Interactions of Tanshinone 2A with SMT (*N. gruberi*)**


Tanshinone 2 A is an abietane diterpenoid which can be found in *Saliva miltorrhiza* and is known to have anti-inflammatory, antineoplastic, cardiac treatment [28,29,30]. It is known to influence Akt-signaling and protect cardiomyocytes against doxorubicin-induced cardiac injury [28,30,31]. One study found that it influences the anti-oxidative capacity of protective enzymes in the heart [30]. It has good bioavailability and can cross the blood-brain-barrier.

The binding affinities for the compound with ERG6, XM, XP are −7.9 kcal/mol, −8.6kcal/mol, and −9.6 kcal/mol, respectively. For XM-Tanshinone 2A, electron cloud interactions like pi-alkyl, pi-anion, and hydrogen bonds are formed with the amino acids Asp280, Phe134, and Asn130. Figure 7g depicts pi-anion interactions between ligand and Asp280 in yellow. Pi-alkyl interaction between ligand and Phe134 in pink. The conventional hydrogen bond between the ligand and Asn130 is shown in green. Figure 7h depicts XM SMT protein surface and Tanshinone 2 A. Tanshinone 2 A targeted methyltransferase (amino acids 118–219) domain and mycolic acid cyclopropane synthetase (amino acids 61–274) domain are for XM. For XP-Tanshinone 2A, electron cloud interactions like pi-alkyl, pi-pi stacked, pi- pi T-shaped, pi-anion, alkyl interactions are formed with amino acids Glu190, Tyr218, Trp284, Trp298. Figure 8g depicts pi-anion interactions between ligand and Glu190 in golden yellow. Pi-Pi stacked interactions between the ligand and Tyr218 and Trp284 are shown in pink. Pi-Pi T-shaped interactions between ligand and Trp284, Trp298 in pink. Pi-Alkyl interactions are shown in purple between the ligand and Trp298, Tyr218. Figure 8h depicts XP SMT protein surface and Tanshinone 2 A. Tanshinone 2 A targeted methyltransferase (amino acids 118–219) domain and the sterol methyltransferase C-terminal (amino acids 298–358) domain for XP.


**Analysis of Docking Interactions of Catharanthine with SMT (*N. gruberi*)**


Catharanthine is a monoterpenoid indole alkaloid produced by *Catharanthus roseus*, *Catharanthus trichophyllus,* and *Tabernaemontana catharinensis* [32,33]. It is a product of medicinal plants and is known to be a bioactive compound. For XM-Catharanthine, electron cloud interactions like pi-alkyl, pi-pi stacked, alkyl interactions and hydrogen bonds are formed with amino acids Tyr221, Phe134, and Val278. Figure 7i depicts the alkyl interaction between the ligand and Val278 is shown in pink. Pi-alkyl interaction between ligand and Try221. Pi-Pi stacked interactions shown in light pink between ligand and Phe134. Catharanthine targeted the methyltransferase (amino acids 118–219) domain for XM. Figure 7j depicts the XM protein surface and Catharanthine. For XP-Catharanthine, electron cloud interactions like Pi-Alkyl, Pi-Pi T-shaped, alkyl interactions and hydrogen bonds are formed with the amino acids Lys104, Ile128, Tyr187, Phe353. Figure 8i depicts Pi-Pi T-shaped interactions between the ligand and Phe353 shown in pink, purple-blue. Hydrogen bond interaction between ligand and Lys104 is shown in green. Alkyl interactions between ligand and Ile128 shown in purple-blue. Pi-Alkyl interaction between ligand and Lys104 and Tyr187 shown in purple-pink. Figure 8j depicts XP SMT protein surface and Catharanthine. Catharanthine targeted the methyltransferase (amino acids 118–219) domain and sterol methyltransferase C-terminal (amino acids 298–358) domain for XP. The compound showed good bioavailability and could cross the blood-brain-barrier.

Each protein, i.e., ERG6, XP, and XM SMTs showed specific binding pockets. Interestingly, the strongest interactions were observed around the same pocket in each protein across various ligands. For the XM protein, the amino acid Phe134 was involved in the binding of four of the top ligands. For the XP protein, Tyr218 and Trp298 were involved in the binding of four of the top ligands, and Phe353 was also involved in multiple ligands. For XM, the binding pocket predictions were consistent with the results of our experiment. The orange binding pocket given in Figure 4g–i and amino acid residues under the orange pocket are given in Appendix A and show that this binding pocket gave the best binding. For XP, the amino acids involved in binding ligands were not involved directly. The amino acids predicted were in close proximity to the ligand as shown by Ligplots for XP in Figure 9b, Figure 10b, Figure 11b and Figure 12a,c. This could indicate that the binding sites are consistent for each protein and further studies could be done by specifically targeting these amino acid regions.


**Analysis of Amino Acid Residues of ERG6 (yeast) & SMT (*N. gruberi*) Interacting with Ligands**


As shown in Table 1, the best ligands for XM vs XP vs ERG6 are different compared to XM vs XP in which only XM and XP ligand binding affinities are taken into account. This can be accounted for by their differences in the structures and amino acid sequences and also because ERG6 has a complete protein model as compared to XM and XP that has partial structures, albeit containing conserved enzymatic domains.

The ligands showing a strong binding to ERG6 but weakly to XM or XP could indicate that these ligands would bind to regions which are not covered in the partial proteins.

ERG6 interacting amino acid residues with Cyclopamine are ASN302, GLY84, PHE100, TRP85, GLY86, SER87, ILE104, GLY132, GU136, ARG135, ARG139 as shown in Figure 9a, while, amino acid residues of XP are LYS104, GLU271, PHE353, ILE275, VAL273, TYR218, SER351, TYR285, TRP298, ILE189, PRO124, TRP284 as shown in Figure 9b. XM interacting amino acid residues are TRP223, ILE317, HIS197, ILE253, PRO251, VAL321, GLY322, ILE244, ASP248, LYS246 as shown in Figure 9c.

While ERG6 interacting amino acid residues with Chelerythrine are TYR83, THR317, LEU341, THR338, PHE80, SER334, LEU31, VAL337, ALA330, MET324, VAL321, MET320, LEU28 as shown in Figure 10a, while, amino acid residues of XP TRP298, GLU190, TRY285, SER351, GLU271, TRP284, VAL273, PHE353, LYS104, TYR218, ILE189 as shown in Figure 10b. XM interacting amino acid residues are ASP110, PHE134, TYR221, TYR,190, ILE131, ASN130, GLU193, PRO127, ILE192, GLN281, ASP280, VAL278, THR278 as shown in Figure 10c.

Similarly, ERG6 interacting amino acid residues with Berberine are ILE22, ILE155, ALA156, TYR160, LYS163, LYS159, GLY23, ARG15, LYS175, ASP152 as shown in Figure 11a, while, amino acid residues of XP TYR187, ILE189, TRP298, TRP284, TYR218, LYS104, GLY216, PHE353, GLN105, ILE128 as shown in Figure 11b. XM interacting amino acid residues are ILE131, PHE134, THR135, ASN130 LYS313, ASP110, LEU111, TYR190, TYR221, VAL221, VAL278, ASP280, GLN281 as shown in Figure 11c.

XP and Tanshinone 2 interacting amino acid residues are as follows: PRO224, ILE189, PHE353, TYR218, TRP284, TYR285, SER248, PHE353, GLU190 as shown in Figure 12a. XM and Tanshinone 2 interacting amino acid residues are as follows: CYS204, GLY121, LEU120, ALA198, HIS145, ASN144, HIS197, LYS199, PHE172, ALA194 as shown in Figure 12b,c. XP and Catharanthine interacting amino acid residues are as follows: ILE128, GLY216, PHE353, GLN105, TYR187, LYS104, TYR187 as shown in Figure 12c, while, XM and Catharanthine interacting amino acid residues are GLN281, THR279, ASP280, TYR221, TYR190, ILE131, PHE134, ASP110, TYR335, VAL278, GLY277 as shown in Figure 12d.

## 3. Discussion

*Naegleria fowleri* is the only pathogenic form of *Naegleria* sp. and exists in both amoeboid and flagellate forms. *N. fowleri* is known to cause PAM, which has a >97% mortality rate. Lipid biosynthesis is a critical pathway for this amoeba and their inhibition has been shown to arrest their proliferation [15]. Previous studies have shown that *N. gruberi* uses lipids as their preferred substrate for energy purposes [17]. Lipid biosynthesis pathways of *N. gruberi* are well established, and recently it was shown that *N. fowleri* has similar genes for lipid metabolism as *N. gruberi* [17].

The ergosterol biosynthesis pathway provides a unique drug target opportunity, as many of the key enzymes involved in this pathway are absent in humans. Sterol 24-C Methyltransferase (SMT) is one such enzyme that is not found in humans.

Sterol 24-C Methyltransferase (SMT) plays a critical role in ergosterol biosynthesis and has been previously predicted to be one of the important drug targets. SMT from *N. gruberi* has two forms that are predicted to be isozymes. Both the enzymes are homologous to ERG6 derived from yeast.

None of the SMTs and ERG6 structures have been solved using high resolution biophysical techniques. Hence, it was imperative to predict the structure. While Alphafold predicted the structure for ERG6, it passed the Ramachandran plot analysis without any errors. SMT isozymes were analyzed using multiple software such as I-TASSER, Phyre 2.0 and Swiss-modeler. The structures predicted by the three types of software were tested using Molprobity and Ramachandran plot analysis. After analysis, only the partial structures for SMT predicted by the Swiss-modeler were used, as the I-TASSER and Phyre 2.0 showed huge errors. The partial structures of SMT contain all of the enzymatic domains.

After the protein structure was optimized for docking, multiple ligands, both natural product derived and synthetic, were tested. Only five natural product derived ligands were selected in the end, as they could cross the BBB.

## 4. Materials and Methods


**Retrieval and preparation of Ligands:**


A large online library from APExBIO was used to select naturally occurring compounds and their derivatives for the purposes of screening. These compounds were identified to be from various sources like plants, fungus etc. Natural compounds and their derivatives are known for their bioactivity and medicinal properties. The compounds were selected and chemical compound databases like PubChem and ChemSpider were used to retrieve the ligands. Certain compounds were semisynthetic. Studies have shown that Semisynthetic compounds may have better drug-likeness properties as compared to natural compounds [34]. The 3D structure of the ligands were downloaded in SDF and MDL format from PubChem (https://pubchem.ncbi.nlm.nih.gov/ (accessed on 31 July 2022) and ChemSpider(http://www.chemspider.com/Default.aspx (accessed on 31 July 2022).

These ligands were then minimized and converted to ligands using the Open babel plug-in of PyRx. The SMILES of the ligands were also saved to be used for ADME studies of SwissADME (http://www.swissadme.ch/ (accessed on 31 July 2022).


**ADME and toxicity assays:**


ADME stands for absorption, distribution, metabolism and excretion. Good ADME and toxicity scores are excellent predictors of a potential drug [18]. In silico ADME and toxicity studies have helped researchers save resources, time and energy. The absorption rate helps determine the dosage and mode of action. Oral bioavailability is preferred as compared to other modes of drug administration. The possibility of a drug to be used or rejected depends on their toxicity. The half-life and stability of a chemical compound determines the frequency of drug administration. Compounds need to have a balance between activity and pharmacokinetic properties to ensure that these give a promising result for in-vivo assays. Distribution and availability of drugs to the correct target is vital for the efficacy of the drug in vivo. A drug that can cross the blood brain barrier has to have certain criteria. Compounds that show a certain level of drug-likeness are deemed as capable of being taken forward from wet lab to clinical trials.

For our study, the main criteria for selection of the compounds for further studies would have been blood-brain-barrier crossing compounds, which would show high absorbance, oral bioavailability, low toxicity and zero to minimal drug-likeness violations. Lipinski’s rule of Five was also tested [19]. These tests help screen out drugs which may fail due to inferior ADME properties.


**Retrieval, modelling and preparation of Protein:**


The protein Sterol 24-C Methyltransferase (SMT) was retrieved from AlphaFold, an AI-based online databank. Alphafold AI can predict structures with higher accuracy than any other software-based method [35]. The lack of an x-ray crystallography structure of *Naegleria fowleri* SMT was addressed by homology modeling using ERG6 protein [(Uniprot: P25087)] expressed in *Saccharomyces cerevisiae* [(strain ATCC 204508 / S288c)]. Since *Naegleria* sp. is a eukaryote, it was necessary to retrieve the protein from a closely related organism.

Major studies related to *Naegleria fowleri* studies are conducted on *Naegleria gruberi* which is a closely related non-pathogenic amoeba and also because it is easy to culture. Due to the lack of SMT amino acid sequence and protein structure from *N. fowleri,* amino acid sequence from SMT derived from *N. gruberi* was used to build SMT structure using online tools such as Swiss-Model, I-TASSER and Phyre2. Of the three amino acid sequences found in gene bank, two sequences were chosen due to their differences for docking and drug analysis. The sequences XP_002671982.1 (XM) and XP_002680047.1 (XP) were sent for model building. 

The structure selection was based on the confidence of alignment coverage, structural analogues, and the Ramachandran plot. The model predicted by the Swiss-modeler model was chosen for docking purposes. The protein structure, though partial, covers all the major domains that have enzymatic activity, according to the NCBI CDD web tool. A literature search revealed that two different sequences were deposited in the NCBI database for the same protein. Both protein structures were built using Swiss-modeler. 

The SMT protein structures for XM and XP were selected based on sequence coverage and GMQE scores for the protein models generated by Swiss-Model [36,37]. The models are evaluated based on two quality measurement factors, namely GMQE and QMEANDisCo (https://swissmodel.expasy.org/docs/help#GMQE (accessed on 31 July 2022). The scores are given between 0 and 1 for both. GMQE stands for global model quality estimate and the score depends on coverage and alignment of the model to the template sequence. If the structure created is not covered completely, the GMQE score drops. GMQE helps in selecting optimal models even before building the model and gets updated by taking QMEANDisCo global score to obtain more reliable quality scores. QMEANDisCo global scores are based on average per-residue QMEANDisCo scores and is estimated for a large set of models. The error estimate is calculated on root mean square difference between QMEANDisCo global score and IDDT, where models similar in size to the template are considered. Table 5 depicts the complete amino acid sequence for the Yeast SMT, XM & XP SMT. The highlighted regions show the regions covered by Swiss-model proteins.

The protein was then prepared for docking using Chimera. Polar hydrogen and charges were added and the missing residues were fixed in Chimera. The protein was saved as a PDB file and then minimized using Chimera. The minimized protein was saved again as a PDB file for docking analysis.


**Virtual Screening and Visualization:**


Sterol 24-C Methyltransferase is an enzyme found in *Naegleria fowleri* which is responsible for lipid biosynthesis, lipid metabolism, steroid biosynthesis, steroid metabolism, sterol biosynthesis, and sterol metabolism [38]. The prepared and minimized SMT protein was tested for the presence of pockets for drug binding using the online website https://proteins.plus/ (accessed on 31 July 2022). The protein was made as a macromolecule/target and 210 ligands were docked against the protein using the PyRx docking tool. PyRx helps in screening large libraries of molecules within short spans of time and is user-friendly [39].

The scoring function is vital for the success of any screening of ligands. Autodock Vina plug-in in PyRx 0.8 gives values such as binding affinity and RMSD scores which assists in predicting the possibility of stronger interactions between each ligand and target.

The autodock Vina plug-in tool was used for the docking studies. The grid box was used to cover the entirety of the protein, where all the likely binding pockets were identified. The binding affinity of the ligand and the protein interactions were saved at the end of the docking process and saved as a CSV file. The interactions were visualized on Discovery Studio visualization tool [40]. Discovery Studio was used as a visualization tool to study various interactions between the ligand and target with different colors. The distance, type of bonds, amino acids involved, and the nature of hydrogen bonds in proximity to the ligand were observed using this software. It gives you the freedom to interact with the protein and improve the visual look of the image by using varying colors, receptor surface, visibility of the ligand/target, and to generate 2D interaction plots. The software gives information on the aromatic surface, H-bond donor/acceptor presence, ionizability, basic/acidic nature, and positive/negative charge of the surface of the target.

## 5. Conclusions

Five ligands, i.e., Cyclopamine, Chelerythrine, Berberine, Tanshinone 2 A, and Catharanthine were identified as potential drug candidates for their ability to inhibit SMT and arrest the growth of *N. fowleri*. XP and XM are the two isozymes of SMT. Hence, this study identified those drug candidates that could inhibit both the isozymes successfully. Apart from the strong binding affinity at the enzymatic domains, other criteria for selecting the five ligands were their ability to permeate BBB, GI absorption, drug-likeness, and solubility. All the five ligands fulfilled the rule of five for absorption and permeation, i.e., they had fewer than five hydrogen bond donors, their molecular weight was less than 500 g/mol, MlogP was less than 4.15 and the number of hydrogen bond acceptors was less than 10. Cyclopamine fulfilled all the criteria except for the MlogP value being slightly higher, which we predict might not be an issue considering it fulfilled all the other criteria such as bioavailability, water solubility, and the four other rules. Chelerythrine, Berberine, Tanshinone 2 A, and Catharanthine fulfilled all the rules of the five and also other pharmacokinetic indicators such as bioavailability, GI absorption, etc. It is worth mentioning that the strongest binding affinity was shown by a ligand named Solamargine. It has the highest binding affinity, but cannot be used due to its high molecular weight and its inability to cross the blood-brain barrier [41]. Since cyclopamine blocks the sonic hedgehog pathway, its potency against various cancers is being investigated. Its two functional analogs, vismodegib, and sonidegib, were already approved by the Food and Drug Administration in 2012 and 2015, respectively. Interestingly, chelerythrine has both anti-bacterial properties as well as apoptotic properties. It can inhibit SERCA and induce apoptosis and is a potent inhibitor of protein kinase C [23,42]. Berberine, Tanshinone 2 A and Catharanthine have also shown anti-cancer like properties [29,43]. It is interesting to note that potential cancer therapeutics could also work as anti-amoebic chemotherapeutics.

Our future work includes testing selected drugs on *N. gruberi* to experimentally determine their efficacy and toxicity. We are also conducting molecular dynamics to better understand the interaction of selected drugs with SMT. Insights from these experiments will help with the selection of drugs to target SMT.

## Figures and Tables

**Figure 1 molecules-27-05727-f001:**
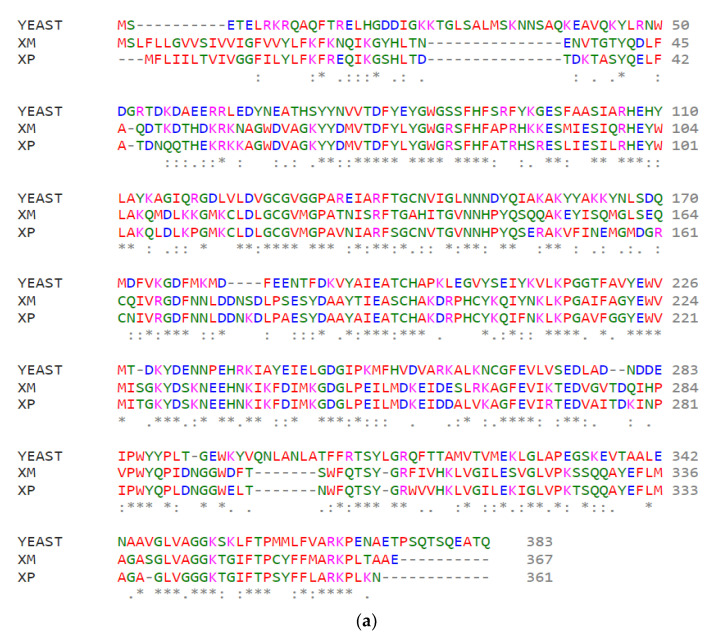
**Sequence comparison of SMT:** (**a**) Comparison between yeast derived ERG6, *N. gruberi* derived XM, and XP. (**b**) Comparison between *N. gruberi* derived XM and XP. Light yellow box denotes the SAM domain while the light orange box denotes the sterol methyltransferase C-terminal domain. The red lines denote the amino acid residues that are predicted to bind S-Adenosylmethionine.

**Figure 2 molecules-27-05727-f002:**
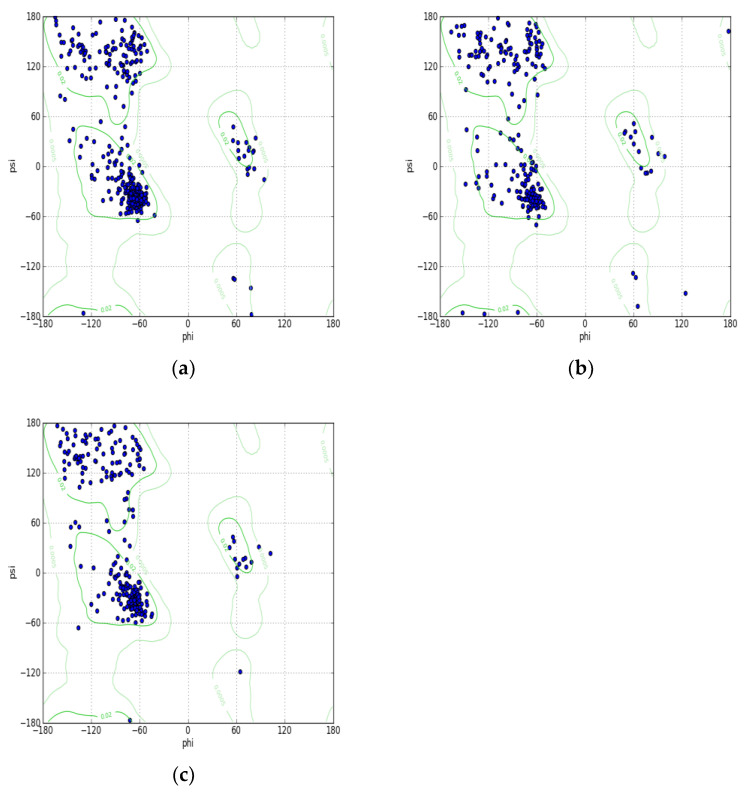
**The Ramachandran plot analysis of the protein structures:** (**a**) The Ramachandran plot for Yeast protein. Highly preferred observations are: 370 (97.113%). Preferred observations are: 10 (2.625%). Questionable observations are: 1 (0.262%). (**b**) The Ramachandran plot for XP protein. Highly Preferred observations are: 244 (95.686%). Preferred observations are: 10 (3.922%). Questionable observations are: 1 (0.392%). (**c**) The Ramachandran plot for XM protein. Highly preferred observations are: 240 (96.386%). Preferred observations are: 8 (3.213%). Questionable observations are: 1 (0.402%).

**Figure 3 molecules-27-05727-f003:**
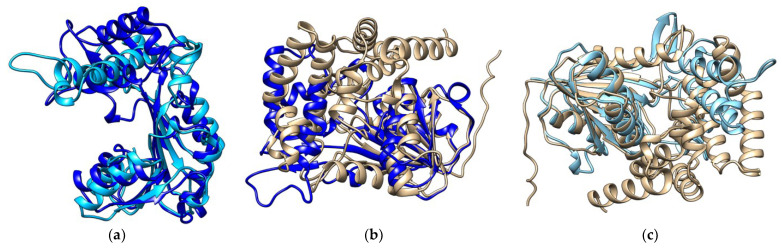
**Structural comparison of the models by superimposition**. (**a**) Superimposed image of protein XP (Light blue) on protein XM (dark blue). (**b**) Superimposed image of proteins ERG6 (YELLOW) and XM (DARK BLUE). (**c**) Superimposed image of proteins ERG6 (Yellow) and protein XP (light blue).

**Figure 4 molecules-27-05727-f004:**
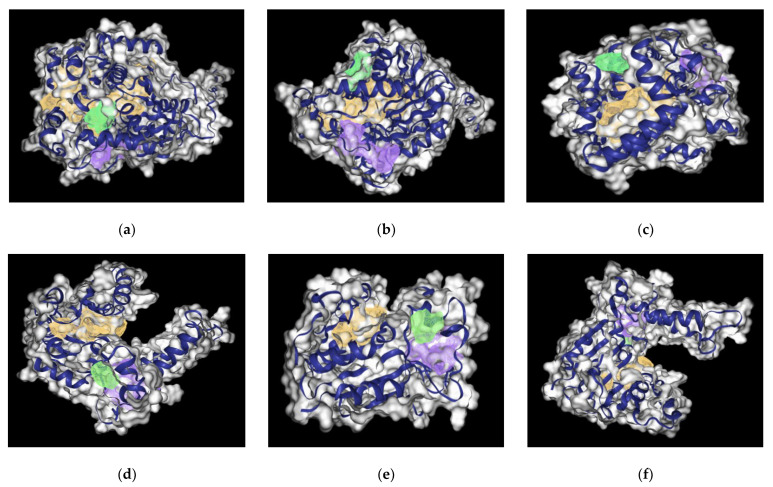
**Binding pocket analysis for modelled proteins**. Images (**a**–**c**) depict the top drug binding pockets within the yeast derived protein ERG6 structure at three different angles rotated at 90° to each other. (**d**–**f**) show *N. gruberi* derived protein XP and (**g**–**i**) protein XM structure at three different angles rotated at 90° to each other. The image shows color with decreasing order of druggability score, with the color coded as orange, purple and green, respectively.

**Figure 5 molecules-27-05727-f005:**
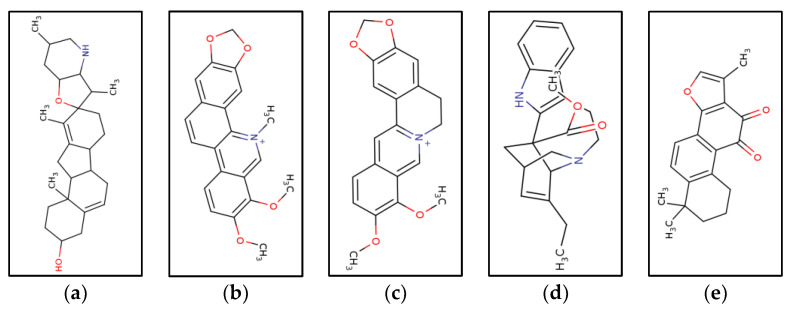
**Chemical structures of the selected ligands.** (**a**) Cyclopamine. (**b**) Chelerythrine. (**c**) Berberine. (**d**) Tanshinone 2A. (**e**) Catharanthine. Structures were prepared in ChemDB Cheminformatics portal (http://cdb.ics.uci.edu/cgibin/Smi2DepictWeb.py (accessed on 31 July 2022).

**Figure 6 molecules-27-05727-f006:**
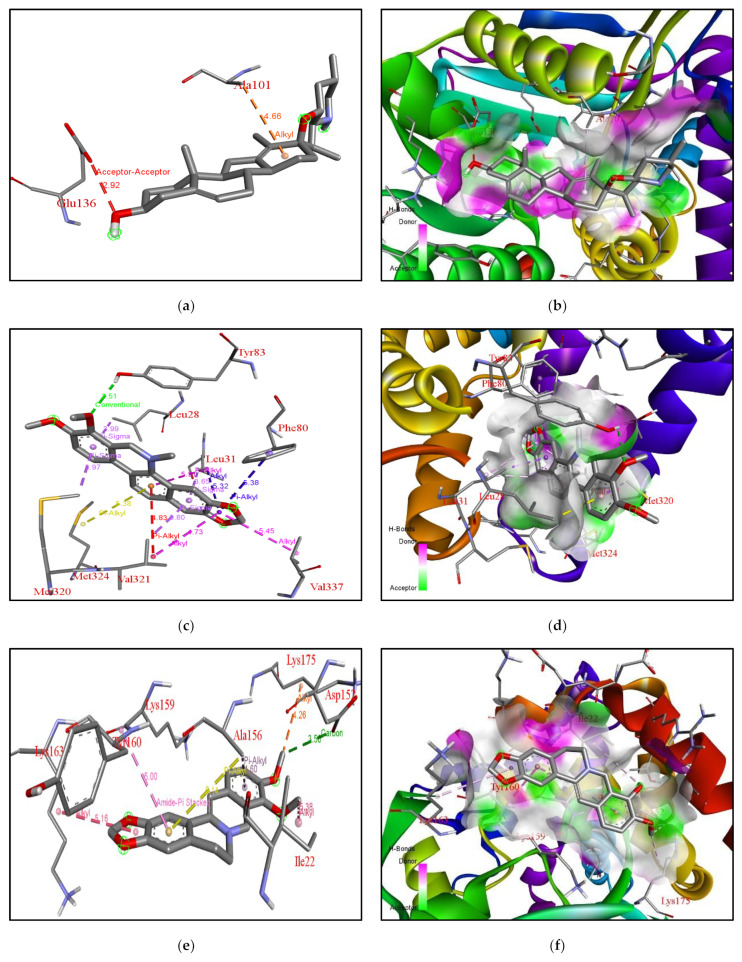
**Analysis of docking pocket of ERG6:** Interactions, bond lengths and hydrogen bond donor and acceptor around ligand binding pocket for yeast ERG6-ligand interaction are shown in these figures. (**a**) ERG6-cyclopamine acceptor-acceptor interactions in red, alkyl interactions in orange, red. (**b**) ERG6 surface at proximity to cyclopamine. (**c**) ERG6-chelerythrine interactions in red, yellow, magenta, blue, purple, pink, green. (**d**) ERG6 protein surface at proximity to Chelerythrine. (**e**) ERG6 -berberine interactions in brown, yellow, pink, red, orange, green. (**f**) ERG6 surface at proximity to berberine colored pink or green based on hydrogen bond donors or acceptors.

**Figure 7 molecules-27-05727-f007:**
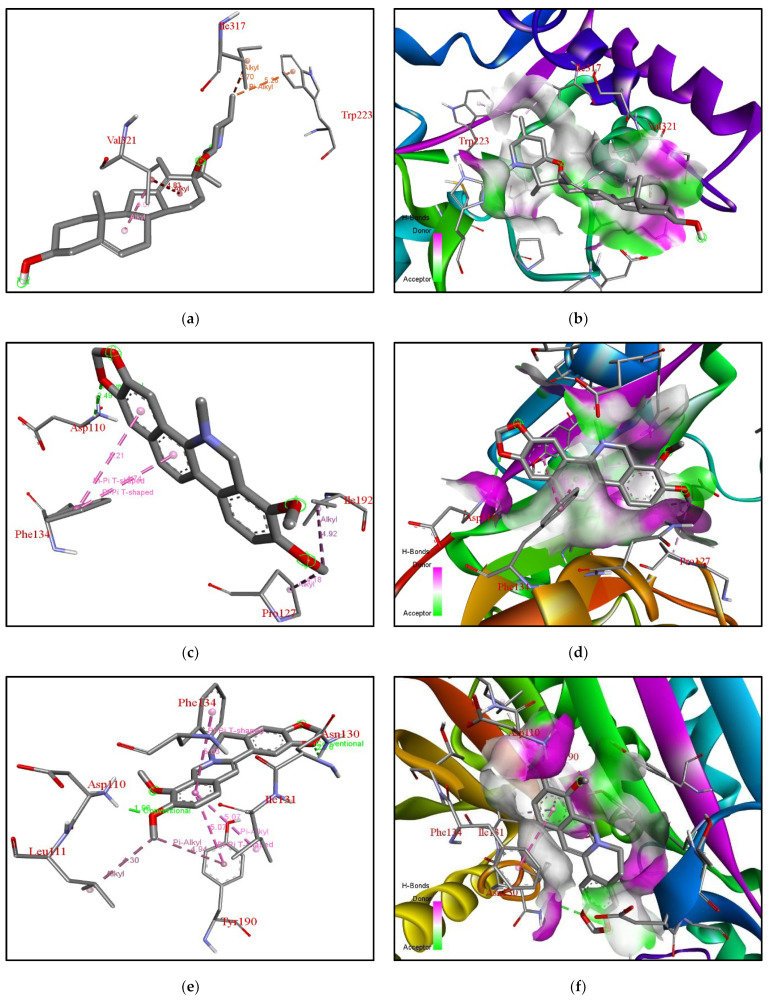
**Analysis of docking pocket of XM-SMT:** Interactions, bond lengths and hydrogen bond donor and acceptor around ligand binding pocket for XM SMT-ligand interaction are shown in these figures. (**a**) XM-cyclopamine interactions in red, pink, orange. (**b**) XM protein surface at proximity to ligand. (**c**) XM-chelerythrine alkyl interactions in pink., green, purple. (**d**) XM SMT protein surface at proximity to Chelerythrine colored pink or green based on Hydrogen bond donors or acceptors. (**e**) XM-berberine Alkyl interaction in metallic pink., green, pink. (**f**) XM SMT surface at proximity to Berberine (**g**) XM-Tanshinone 2 A interactions in yellow, pink, and green. (**h**) XM surface at proximity to Tanshinone 2 A pink or green based on Hydrogen bond donors or acceptors. (**i**) XM-catharanthine interactions in pink. (**j**) XM protein surface at proximity to Catharanthine pink or green based on hydrogen bond donors or acceptors.

**Figure 8 molecules-27-05727-f008:**
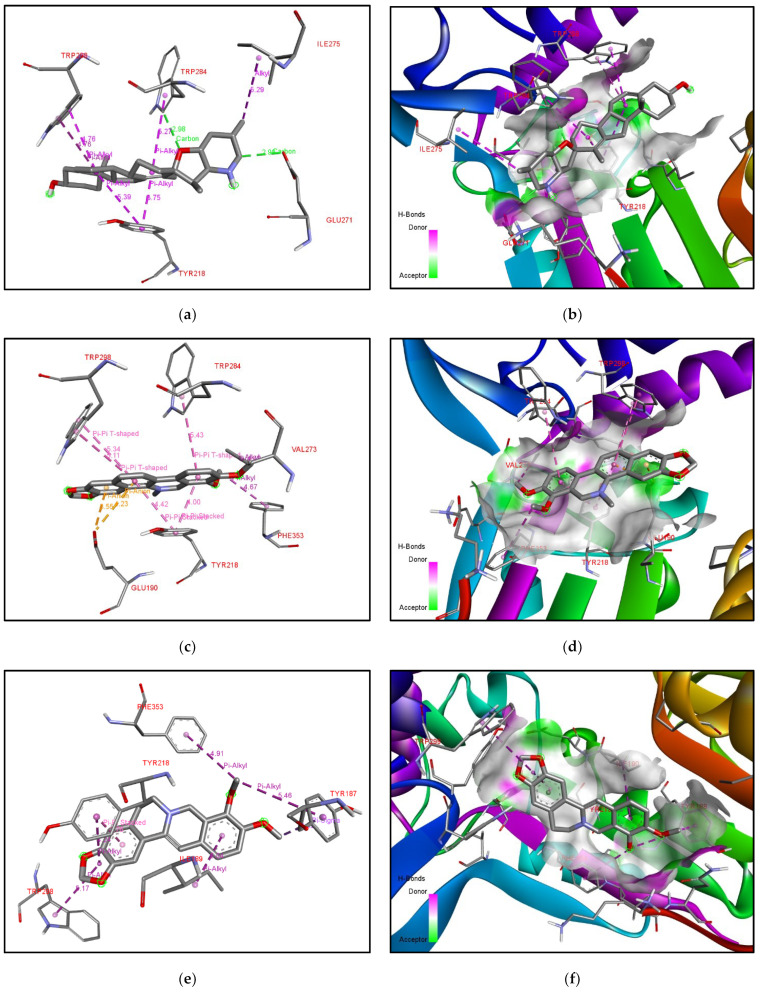
**Analysis of docking pocket of XP-SMT:** Interactions, bond lengths and hydrogen bond donor and acceptor around ligand binding pocket for XP SMT-ligand interaction are shown in these figures. (**a**) XP-cyclopamine interactions in purple and green. (**b**) XP protein surface at proximity to Cyclopamine colored pink or green based on Hydrogen bond donors or acceptors. (**c**) XP-Chelerythrine interactions in golden yellow, pink, purple. (**d**) XP protein surface at proximity to Chelerythrine colored pink or green based on Hydrogen bond donors or acceptors. (**e**) XP-Berberine interactions in pink and purple. (**f**) XP protein surface at proximity to Berberine colored pink or green based on Hydrogen bond donors or acceptors. (**g**) XP-Tanshinone 2 A interactions in golden yellow, pink and purple. (**h**) XP protein surface at proximity to Tanshinone 2A colored pink or green. (**i**) XP-Catharanthine interactions in pink, purple-blue, green, pink, and blue. (**j**) XP protein surface at proximity to Catharanthine colored pink or green based on hydrogen bond donors or acceptors.

**Figure 9 molecules-27-05727-f009:**
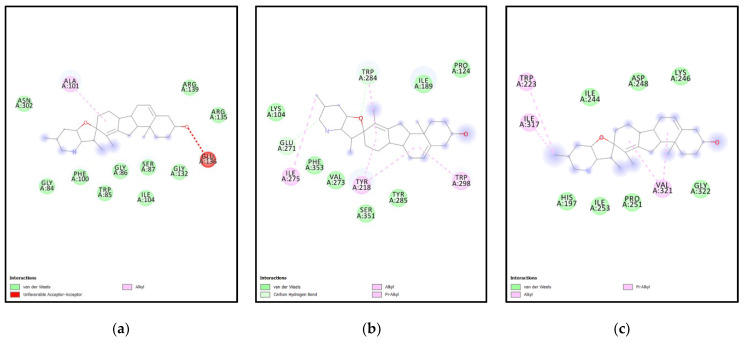
**Ligplots showing amino acid residues involved in****Cyclopamine-protein interactions** (**a**) ERG6-Cyclopamine interacting amino acid residues (**b**) XP-Cyclopamine interacting amino acid residues (**c**) XM-Cyclopamine interacting amino acid residues.

**Figure 10 molecules-27-05727-f010:**
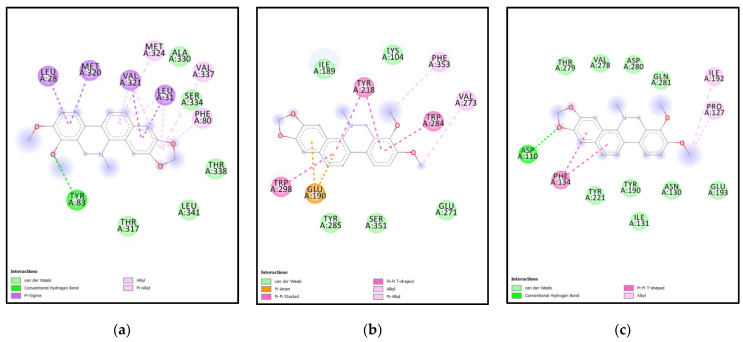
**Ligplot showing amino acid residues involved in****Chelerythrine -protein interaction for** (**a**) Yeast-Chelerythrine interacting amino acid residues (**b**) XP-Chelerythrine interacting amino acid residues (**c**) XM-Chelerythrine interacting amino acid residues.

**Figure 11 molecules-27-05727-f011:**
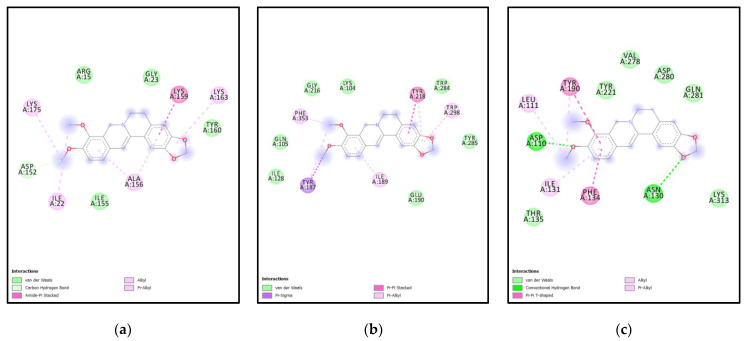
**Ligplot showing amino acid residues involved in ligand-protein interaction for** (**a**) Yeast-Berberine interacting amino acid residues (**b**) XP-Berberine interacting amino acid residues (**c**) XM-Berberine interacting amino acid residues.

**Figure 12 molecules-27-05727-f012:**
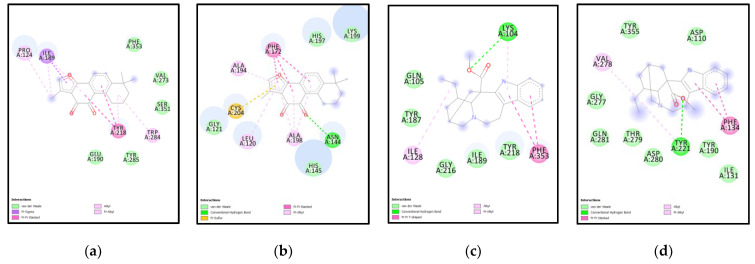
**Ligplot showing amino acid residues involved in ligand-protein interactions for XP & XM** (**a**) XP-Tanshinone 2 interacting amino acid residues (**b**) XM-Tanshinone 2 interacting amino acid residues (**c**) XP-Catharanthine interacting amino acid residues (**d**) XM-Catharanthine interacting amino acid residues.

**Table 1 molecules-27-05727-t001:** Comparative analysis of binding affinities of ligands against XM, XP, & Yeast proteins.

Rank	XM vs. XP vs. Yeast	Binding Affinities (XM, XP, YEAST) Kcal/mol	XM vs. XP	Binding Affinities (XM, XP) Kcal/mol
1.	Solamargine (PCID:73611)	−9.4, −10.6, −9.8	Solamargine (PCID:73611)	−9.4, −10.6
2.	Wilforlide A (PCID:158477)	−9, −10.1, −8.4	Obacunone (PCID:119041)	−9.2, −10.5
3.	Obacunone (PCID:119041)	−9.2, −10.5, −8.1	Wilforlide A (PCID:158477)	−9, −10.1
4.	Cyclopamine (PCID:442972)	−8.6, −10.3, −8	Cyclopamine [(PCID:442972)	−8.6, −10.3
5.	Ginsenoside Ro (PCID:11815492)	−8.5, −9.5, −8.5	Tanshinone 2 (PCID:164676)	−8.6, −9.6
6.	Estradiol cypionate (PCID:9403)	−8.5, −9.6, −8	Ginsinoside Ro (PCID:11815492)	−8.5, −9.5
7.	Natamycin (PCID:5284447)	−8.5, −9.4, −8.3	Estradiol Cypionate (PCID:9403)	−8.5, −9.6
8.	Tiliroside (PCID:5320686)	−8.7, −8.3, −8.1	Camptothecin (PCID:2360)	−8.6, −9
9.	Etoposide (PCID:36462)	−8.2, −9.1, −7.9	Natamycin (PCID:5284447)	−8.5, −9.4
10.	Myricitrin (PCID:5281673)	−8, −8.1, −8.5	Epothilone A	−8.3, −9.1
11.	Tigecycline (PCID:54686904)	−8, −8, −8.2	Etoposide (PCID:36462)	−8.2, −9.1
12.	Chelerythrine (PCID:2703)	−7.8, −7.6, −7.9	Timosaponin B 2 (PCID:44575945)	−8.3, −8.7
13.	Luteolin (PCID:5280445)	−7.9, −8.6, −7.2	Tiliroside (PCID:5320686)	−8.7, −8.3
14.	Berberine (PCID:2353)	−7.8, −8.3, −7.3	Beta−carotene (PCID:5280489)	−8.5, −8.3
15.	Butein (PCID:5281222)	−6.6, −6.7, −8	Catharanthine (PCID:5458190)	−8, −8.4
16.			Clotrimazole (PCID:2812)	−8, −8.9

**Table 2 molecules-27-05727-t002:** Top ligands with high druggability and BBB crossing.

Rank	XM vs. XP vs. Yeast	XM vs. XP
1.	Cyclopamine (PCID:442972)	Cyclopamine (PCID:442972)
2.	Chelerythrine (PCID:2703)	Tanshinone 2 (PCID:164676)
3.	Berberine (PCID:2353)	Catharanthine (PCID:5458190)

**Table 3 molecules-27-05727-t003:** Drug likeness and ADME results for Top ligands.

Ligand	Molecular Weight (g/mol)	Lipophilicity (Log P_o/w_ (XLOGP3))	Hydrogen Bond Acceptors, Donors	Water Solubility (Log S (ESOL))
Cyclopamine	411.62	3.52	3, 2	−4.61, Moderately soluble
Chelerythrine	348.37	4.58	4, 0	−5.27, Poorly soluble
Berberine	336.36	3.62	4, 0	−4.55, Moderately soluble
Tanshinone 2 A	294.34	4.33	3, 0	−4.76, Poorly soluble
Catharanthine	336.43	2.80	3, 1	−3.76, Moderately soluble

**Table 4 molecules-27-05727-t004:** Pharmacokinetic properties of top ligands.

Ligand	Druglikeness Violations (Lipinski, Ghose, Veber, Egan, Muegge)	Bioavailability	Blood-Brain Barrier Crossing	GI Absorption	Skin Permeability (cm/s)
Cyclopamine	Lipinski (1, MLOGP > 4.15), Ghose (1, atoms > 70) violations	0.55	Yes	High	−6.31
Chelerythrine	No	0.55	Yes	High	5.17
Berberine	No	0.55	Yes	High	−5.78
Tanshinone 2	No	0.55	Yes	High	−5.02
Catharanthine	No	0.55	Yes	High	−6.36

**Table 5 molecules-27-05727-t005:** Amino acid sequence for complete structure of Yeast SMT protein form Alphafold.

Protein	Amino Acid Sequence
YEAST SMT	MSETELRKRQAQFTRELHGDDIGKKTGLSALMSKNNSAQKEAV QKYLRNWDGRTDKDAEERRLEDYNEATHSYYNVVTDFYEYGW GSSFHFSRFYKGESFAASIARHEHYLAYKAGIQRGDLVLDVGCGV GGPAREIARFTGCNVIGLNNNDYQIAKAKYYAKKYNLSDQMDF VKGDFMKMDFEENTFDKVYAIEATCHAPKLEGVYSEIYKVLKPG GTFAVYEWVMTDKYDENNPEHRKIAYEIELGDGIPKMFHVDVA RKALKNCGFEVLVSEDLADNDDEIPWYYPLTGEWKYVQNLANL ATFFRTSYLGRQFTTAMVTVMEKLGLAPEGSKEVTAALENAAVG LVAGGKSKLFTPMMLFVARKPENAETPSQTSQEATQ
XM complete sequence *	MSLFLLGVVSIVVIGFVVYLFKFKNQIKGYHLTNENVTGTYQDLF AQDTKDTHDKRKNAGWDVAGKYYDMVTDFYLYGWGRSFHFA PRHKKESMIESIQRHEYWLAKQMDLKKGMKCLDLGCGVMGPA TNISRFTGAHITGVNNHPYQSQQAKEYISQMGLSEQCQIVRGDFN NLDDNSDLPSESYDAAYTIEASCHAKDRPHCYKQIYNKLKPGAIF AGYEWVMISGKYDSKNEEHNKIKFDIMKGDGLPEILMDKEIDESL RKAGFEVIKTEDVGVTDQIHPVPWYQPIDNGGWDFTSWFQTSYG RFIVHKLVGILESVGLVPKSSQQAYEFLMAGASGLVAGGKTGIFTP CYFFMARKPLTAAE
XP complete sequence *	MFLIILTVIVGGFILYLFKFREQIKGSHLTDTDKTASYQELFATDNQ QTHEKRKKAGWDVAGKYYDMVTDFYLYGWGRSFHFATRHSRE SLIESILRHEYWLAKQLDLKPGMKCLDLGCGVMGPAVNIARFSG CNVTGVNNHPYQSERAKVFINEMGMDGRCNIVRGDFNNLDDN KDLPAESYDAAYAIEATCHAKDRPHCYKQIFNKLKPGAVFGGYE WVMITGKYDSKNEEHNKIKFDIMKGDGLPEILMDKEIDDALVKA GFEVIRTEDVAITDKINPIPWYQPLDNGGWELTNWFQTSYGRWV VHKLVGILEKIGLVPKTSQQAYEFLMAGAGLVGGGKTGIFTPSYF FLARKPLKN

* Highlighted amino acid residues in red denote the partial structures of modelled SMT isozymes XP and XM.

## Data Availability

Data is contained within the article and Appendix A.

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
