# Peer review of "Virtual Screening of Alkaloid and Terpenoid Inhibitors of SMT Expressed in Naegleria sp."

_molecules, 2022, doi:10.3390/molecules27175727_

Round 1

Reviewer 1 Report

Dear Authors

Greetings

Please find the final Review Report of the Manuscript

Manuscript No.:

1817698

Manuscript title

Virtual Screening of Natural and Synthetic Inhibitors of Sterol-24 C Methyl transferase Enzyme Derived from Naegleria sp.

Article type:

Research Article

Journal:

Molecules

Corresponding Author:

Supriyo Ray

Submitted on:

Manuscript submitted on 07.07.22

I have gone through the manuscript entitled “Virtual Screening of Natural and Synthetic Inhibitors of Sterol-24 C Methyl Transferase Enzyme Derived from Naegleria sp.. The article presents the importance of Sterol 24-C Methyl transferase, an enzyme involved in ergosterol biosynthesis that could serve as an important drug target for N. fowleri. The research article comes up with an appropriate introduction followed by results, discussion, materials and methods that are written in a very accurate sequential manner. The article first highlighted the importance of Sterol 24-C Methyltransferase in ergosterol biosynthesis and then moved on to screen its potential inhibitors and their derivatives having significant binding affinity with the target. But while reviewing this article I felt some suggestions that the authors need to consider for the improvement of this research article. Some points are listed below:

1.     The introduction section should highlight the role of sterol-24 C Methyltransferase inhibitors and their significance in N. fowleri.

2.     The author should mention the rationale for sterols as targets in N. fowleri.

3.     The methodology could have been explained in a much better way. For instance, Virtual Screening and Visualization section should be modified as per the text.

4.     The authors could have extended their study further, by giving a touch of MD simulation that could increase the weightage of the article. The authors can consider the following articles for the same:

a.     https://doi.org/10.1007/s12032-022-01748-2,

b.     https://doi.org/10.1038/s41598-022-10796-7

5.     The discussion is very well-sequenced according to the text.

6.     The authors should add a conclusion section to the text, in order to get the crux of the study.

7.     The manuscript needs to be edited for English and grammatical errors.

In order, to improve the quality of the manuscript, it would be beneficial if the manuscript is once again checked for the suggestions provided above. The manuscript needs a major revision before it is considered for publication.

Regards 

Author Response

I have gone through the manuscript entitled “Virtual Screening of Natural and Synthetic Inhibitors of Sterol-24 C Methyl Transferase Enzyme Derived from Naegleria sp.”. The article presents the importance of Sterol 24-C Methyl transferase, an enzyme involved in ergosterol biosynthesis that could serve as an important drug target for N. fowleri. The research article comes up with an appropriate introduction followed by results, discussion, materials and methods that are written in a very accurate sequential manner. The article first highlighted the importance of Sterol 24-C Methyltransferase in ergosterol biosynthesis and then moved on to screen its potential inhibitors and their derivatives having significant binding affinity with the target. But while reviewing this article I felt some suggestions that the authors need to consider for the improvement of this research article. Some points are listed below:

  1. The introduction section should highlight the role of sterol-24 C Methyltransferase inhibitors and their significance in N. fowleri.

Thank you very much for your valuable comments and for taking your valuable time to review our paper. We have made all the changes as recommended.

The role of inhibitors are explained under their respective sections. It has not been included under introduction as it was a result of rigorous screening and they are experimental outcome. Hence, including it in the introduction will be misleading as it would mean that we already knew about the inhibitors even before we did the experiment and those were the only ones that were studied.

  1. The author should mention the rationale for sterols as targets in N. fowleri.

Thanks again for your comments. We have elaborated as suggested in the introduction.

  1. The methodology could have been explained in a much better way. For instance, Virtual Screening and Visualization section should be modified as per the text

Thanks again for your feedback. We have elaborated and improved those sections.

  1. The authors could have extended their study further, by giving a touch of MD simulation that could increase the weightage of the article. The authors can consider the following articles for the same:
  2. https://doi.org/10.1007/s12032-022-01748-2,
  3. https://doi.org/10.1038/s41598-022-10796-

Thank you very much for your constructive feedback. We have started doing the MD simulations and is still continuing. The amount of data it has generated is so big that it is beyond the scope of this manuscript. We are working on another manuscript that will discuss these MD simulations. But again thank you very much for your comments as this was very helpful.

  1. The discussion is very well-sequenced according to the text

     Thanks a lot for your kind comments.

  1. The authors should add a conclusion section to the text, in order to get the crux of the study

       Thanks a lot again for your insightful comments. We have added the conclusion section summarizing the outcome.

  1. The manuscript needs to be edited for English and grammatical errors.

    Thanks a lot again for your comments. We have extensively revised the manuscript for the language errors.

Reviewer 2 Report

Manuscript is well written and concise data set was collected by author to support the findings. Introduction covers all basic information surrounding the topic. But there is need to address some points such as abstract is not properly written as it lack major findings of study.  Results were explained wisely by referring figures and tables but discussed in detail in this section. Explain the results thoroughly in discussion section. Conclude the study by summarizing the findings.

Other comments

Review title: “Virtual Screening of Natural and Synthetic Inhibitors of Sterol-24 C Methyltransferase Enzyme Derived from Naegleria sp.” As the Study encompass the screening of natural inhibitors and their derivatives not the synthetic inhibitors.

Headings were not according to format, please review.

Review line# 714.

Ergosterol biosynthesis pathway should be added from where Sterol 24-C-methyltransferase being inhibited

Review line #113- 114, text is highlighted

Review legend under figure #3  (b), (c)

Review heading “Drug docking and binding analysis” as no results were mentioned in it with reference to table.

Favourable and unfavourable regions should be included in Ramachandran plot

Molecular dynamics simulation should be conducted to examine the stability of the homology modeled protein and the ligand-enzyme complex

Conclusion is missing

Future perspective should be included

There are some corrections needed to be done as per formatting

Add proper citation of article in citation box

Line numbers were not properly assigned

Assign numbers to headings and subheadings

Remove extra space between paragraphs

Follow the reference style of journal “International journal of molecular science”

Author can discuss following points in introduction

Why there is need inhibit Sec61 protein e.g. write possible toxicity associated with its over expression

Methodology

Author can discuss the toxicity evaluation of designed analogues

Do present the visualization in in-silico molecular modeling e.g. which interaction takes part in binding of ligand with protein

Author could discuss the adsorption distribution metabolism excretion toxicity analysis over designed analogues so can predict the toxicity.

There are some corrections needed to be done as per formatting

Author names were not formatted according to journal requirement.

Add citation proper citation of article in citation box

Line numbers were not properly assigned

Quality of figure 2 needs to be improved

Remove extra space in line no 61

Remove extra word “10 mM” in line number 66

Author Response

Manuscript is well written and concise data set was collected by author to support the findings. Introduction covers all basic information surrounding the topic.

But there is need to address some points such as abstract is not properly written as it lack major findings of study. Results were explained wisely by referring figures and tables but discussed in detail in this section. Explain the results thoroughly in discussion section. Conclude the study by summarizing the findings.

Thank you very much for your valuable comments and for taking your valuable to review our paper. We have made all the changes as recommended. Thank you again for your suggestions.

Other comments

Review title: “Virtual Screening of Natural and Synthetic Inhibitors of Sterol-24 C Methyltransferase Enzyme Derived from Naegleria sp.” As the Study encompass the screening of natural inhibitors and their derivatives not the synthetic inhibitors.

Thank you very much for your comments. We have revised the title as per your suggestion.

Headings were not according to format, please review.

Thank you very much for your suggestion. It’s now fixed.

Review line# 714.

Thank you very much for your suggestion. It’s now fixed.

Ergosterol biosynthesis pathway should be added from where Sterol 24-C-methyltransferase being inhibited

Thank you very much for your suggestion. It’s now been added to the introduction.

Review line #113- 114, text is highlighted

Thanks a lot for your catching that typo. It’s now fixed.

Review legend under figure #3 (b), (c)

Thanks again for your comments. It’s now fixed.

Review heading “Drug docking and binding analysis” as no results were mentioned in it with reference to table.

Thanks a lot for your comments. It’s now fixed.

Favourable and unfavourable regions should be included in Ramachandran plot

Thanks a lot for your comments. Under the heading Protein Structure & Analysis we had discussed in minutest detail. We have revisited and ensured the accuracy of the data again.

Molecular dynamics simulation should be conducted to examine the stability of the homology modeled protein and the ligand-enzyme complex

Thank you very much for your constructive feedback. We have started doing the MD simulations and is still continuing. The amount of data it has generated is so big that it is beyond the scope of this manuscript. We are working on another manuscript that will discuss these MD simulations. But again thank you very much for your comments as this was very helpful.

Conclusion is missing

Thanks again for your suggestion. It’s now added in the manuscript after the discussion.

Future perspective should be included

Thanks a lot for your comments. It’s now now added in the last few sentences of the conclusion.

There are some corrections needed to be done as per formatting: Add proper citation of article in citation box; Line numbers were not properly assigned; Assign numbers to headings and subheadings; Remove extra space between paragraphs; Follow the reference style of journal “International journal of molecular science”

Thanks again for your sugegstion and comments. We have already worked on it and if there are any more changes required then we will work with the copy editor once the manuscript is accepted to get them fixed. Thanks.

Author can discuss following points in introduction

Why there is need inhibit Sec61 protein e.g. write possible toxicity associated with its over expression

Thanks for your comments but we have not discussed Sec61 protein anywhere in our manuscript.

Methodology

Author can discuss the toxicity evaluation of designed analogues

Do present the visualization in in-silico molecular modeling e.g. which interaction takes part in binding of ligand with protein

Thanks for your comments but we have discussed molecular modeling in ultra detail in figures 7-13.

Author could discuss the adsorption distribution metabolism excretion toxicity analysis over designed analogues so can predict the toxicity.

Thanks for your comments but we have discussed it in Table 10 and 11. In fact, that is how we narrowed down on our drug selection.

There are some corrections needed to be done as per formatting: Author names were not formatted according to journal requirement; Add citation proper citation of article in citation box; Line numbers were not properly assigned

Thanks again for your sugegstion and comments. We have already worked on it and if there are any more changes required then we will work with the copy editor once the manuscript is accepted to get them fixed. Thanks.

Quality of figure 2 needs to be improved

Thanks for your suggestion. The images have been revised and improved.

Remove extra space in line no 61 & Remove extra word “10 mM” in line number 66

Thanks again for your comments. We revised our manuscript for extra space and word even though we did not find any extra space or word in line 61 and 66.

Round 2

Reviewer 2 Report

Authors have improved the manuscript significantly. Therefore, I recommend it for publication.